# Discovering Venom-Derived Drug Candidates Using Differential Gene Expression

**DOI:** 10.3390/toxins15070451

**Published:** 2023-07-09

**Authors:** Joseph D. Romano, Hai Li, Tanya Napolitano, Ronald Realubit, Charles Karan, Mandë Holford, Nicholas P. Tatonetti

**Affiliations:** 1Department of Biostatistics, Epidemiology and Informatics, University of Pennsylvania, Philadelphia, PA 19104, USA; joseph.romano@pennmedicine.upenn.edu; 2Institute for Biomedical Informatics, University of Pennsylvania, Philadelphia, PA 19104, USA; 3Center of Excellence in Environmental Toxicology, University of Pennsylvania, Philadelphia, PA 19104, USA; 4Department of Systems Biology, Columbia University, New York, NY 10032, USA; hl2350@columbia.edu (H.L.); realubitr@gmail.com (R.R.); ck2389@columbia.edu (C.K.); 5Columbia Genome Center, Columbia University, New York, NY 10032, USA; 6Department of Chemistry, CUNY Hunter College, New York, NY 10032, USAmholford@hunter.cuny.edu (M.H.); 7The PhD Program in Biochemistry, Graduate Center of the City University of New York, New York, NY 10016, USA; 8The PhD Program in Chemistry, Graduate Center of the City University of New York, New York, NY 10016, USA; 9The PhD Program in Biology, Graduate Center of the City University of New York, New York, NY 10016, USA; 10Department of Invertebrate Zoology, The American Museum of Natural History, New York, NY 10032, USA; 11Department of Computational Biomedicine, Cedars-Sinai Medical Center, Los Angeles, CA 90069, USA

**Keywords:** venoms, transcriptomics, RNA-Seq, translational bioinformatics, systems biology, drug discovery

## Abstract

Venoms are a diverse and complex group of natural toxins that have been adapted to treat many types of human disease, but rigorous computational approaches for discovering new therapeutic activities are scarce. We have designed and validated a new platform—named VenomSeq—to systematically identify putative associations between venoms and drugs/diseases via high-throughput transcriptomics and perturbational differential gene expression analysis. In this study, we describe the architecture of VenomSeq and its evaluation using the crude venoms from 25 diverse animal species and 9 purified teretoxin peptides. By integrating comparisons to public repositories of differential expression, associations between regulatory networks and disease, and existing knowledge of venom activity, we provide a number of new therapeutic hypotheses linking venoms to human diseases supported by multiple layers of preliminary evidence.

## 1. Introduction

Venoms are complex mixtures of organic macromolecules and inorganic cofactors that are used for both predatory and defensive purposes. Since the dawn of recorded history, humans have exploited venoms and venom components for treating a wide array of illnesses and conditions, a trend which has continued into modern times [1]. Currently, approximately 20 venom-derived drugs are in use worldwide, with 6 approved by the US Food and Drug Administration for clinical use, and many more currently undergoing clinical trials [2]. As new discovery of synthetic small-molecule drugs has slowed considerably in recent decades, venoms and other natural products hold great promise for discovering innovative treatments for disease and injury, especially for diseases that have evaded treatment through conventional medical science.

Depending on the species, a single venom can contain hundreds of distinct compounds [3]. Current estimates suggest that approximately 200,000 venomous animals exist across the tree of life. As a result, venom-derived compounds are an immense library of candidates for drug discovery which are virtually guaranteed by natural selection to be biologically active [4,5]. Venoms and venom-derived compounds that have already been approved for therapeutic use tend to be relatively small peptides (e.g., fewer than ∼20 residues) or smaller analogs that mimic the active effect of peptides [6]. Although most drugs on the market are small molecules, this is due partially to their convenience in drug design, and as novel drug discovery becomes more challenging, the use of larger, biologically derived molecules (named biologics) is becoming increasingly common [7]. Important challenges and considerations in venom-based drug design include maintaining structural stability and ensuring that the compound can be delivered successfully [8].

Toxinologists have applied modern high-throughput sequencing (HTS) methodologies to the study of venoms (a field that has come to be known as venomics) [5]. Venomics generally involves the sequencing and structural identification of multiple types of macromolecules—genomic DNA, venom gland mRNA transcripts, and/or venom proteins—to best evaluate which genes, transcripts, and polypeptides (including post-translational modifications) are present in a venom and responsible for its activity.

Venomics has become a popular framework for drug discovery in recent years. Innovative advances in this area include engineering bacteria to express venom-based drug screening libraries [9], designing high-throughput venom assays for specific drug targets [10], and optimizing venom-derived lead compounds to function better as drugs [11], among others. However, other applications of HTS and biomedical data science beyond the discovery/evaluation of venom components can be used for drug discovery. One such application is the data-driven analysis of perturbational gene expression data, in which human cells are exposed in vitro to controlled dosages of candidate compounds and then profiled for differential gene expression via RNA sequencing (RNA-Seq). Here, we present VenomSeq—a new informatics workflow for discovering associations between venoms and therapeutic avenues of treatment for disease.

Briefly, VenomSeq involves exposing human cells to dilute venoms and then generating differential expression profiles for each venom, comprising the significantly up- and downregulated genes in cells perturbed by the venom. We then compare the differential expression profiles to data from public compendia of perturbational gene expression data to propose new therapeutic hypotheses based on similarities to existing pharmaceutical drugs, and support our findings by validating the differential expression data against gene regulatory networks corresponding to disease states. VenomSeq works in the absence of any predefined hypotheses, instead allowing the data to suggest hypotheses that can then be explored comprehensively using rigorous traditional approaches.

In this paper, we describe the implementation of VenomSeq and apply it to the crude venoms of 25 diverse animal species and 9 purified teretoxins. In doing so, we identified six crude venoms and one teretoxin with strong similarities to publicly available expression profiles for established drug classes. We also provide technical validation in the form of (a) applying VenomSeq to 37 (non-venom) drugs with known activities and (b) previously generated reference data for 19,811 small-molecule compounds, finding that VenomSeq’s therapeutic hypotheses on these data are enriched for drugs’ known therapeutic classes. VenomSeq and all resulting data from our analyses are open source and freely available to the scientific community.

## 2. Results and Discussion

### 2.1. Venom Dosages

In order to optimize the exposure concentrations of each venom, we performed growth inhibition assays on human cells exposed to varying concentrations of the venoms. This is necessary to minimize the impact of toxicity while ensuring the venom is in high enough concentration to exert an effect on the human cells. Since each venom comprises many (largely unknown) molecular components, we performed the assays on samples of venom measured in mass per volume, rather than compound concentration (molarity). We used ${\mathrm{GI}}_{20}$—the concentration of a venom at which it inhibits growth of human cells by 20%—as the effective treatment dose in all subsequent experiments. It should be noted that this does not reflect a desire to select venoms with an inhibitory effect on cell growth. Rather, it is a heuristic approach to establishing effective dosages for high-throughput screening where preferred dosages are not available a priori.

The experimental ${\mathrm{GI}}_{20}$ values and complete dose–response data for each of the 25 venoms are provided in Table A1, a sample of which is reproduced (for *S. maurus*) in Table 1. The resulting growth inhibition curves for all venoms are shown in Figure 1. Venoms from *L. colubrina*, *D. polylepis*, *S. verrucosa*, *S. horrida*, *C. marmoreus*, *O. macropus*, and *P. volitans* did not demonstrate substantial growth inhibition at any tested concentration, so for those venoms, we instead performed sequencing at 1.0 μg μL^−1^, which is the highest concentration used in the growth inhibition curves.

### 2.2. mRNA Sequencing of Venom-Perturbed Human Cells

After determining appropriate dose concentrations for each venom, we performed RNA-Seq on human IMR-32 cells exposed to the individual venoms. Table 2 summarizes the experimental conditions used for sequencing. After transforming the raw sequencing reads to gene counts (see Section 4.5), we compiled the results into a matrix, where rows represent genes, columns represent samples, and cells represent counts of a gene in a sample. For detailed quality control data, refer to Appendix B, which includes links to related files. The raw (i.e., FASTQ files produced by the sequencer) and processed (i.e., gene counts per sample) data files are available for download and reuse on NCBI’s Gene Expression Omnibus database; accession GSE126575.

### 2.3. Differential Expression Profiles of Venom-Perturbed Human
Cells

We constructed differential expression signatures for each of the 25 venoms as described in Section 4.6, where each signature consists of a list (length ≥0) of significantly upregulated genes and a list (length ≥0) of significantly downregulated genes. The specific expression signatures are available on FigShare at https://doi.org/10.6084/m9.figshare.7609160 (accessed on 6 July 2023)). An excerpt from the expression signature for *O. macropus* is shown in Table 3. The total number of differentially expressed genes for each venom ranges from 2 genes (*Laticauda colubrina* and *Dendroaspis polylepis polylepis*) to 1494 genes (*Synanceia verrucosa*). Gene-wise statistical significance is a function of both $\log_{2}$-fold change and the number of observed counts.

Using publicly available differential expression profiles for existing drugs—many with known effects and/or disease associations—we were able to identify statistically significant associations between venoms and classes of drugs. These associations are based on the methods developed by the Connectivity Map (CMap) project [12], and utilize their perturbational differential expression data as the “gold standard” against which to evaluate the venom expression data. In short, this approach uses a Kolmogorov–Smirnov-like signed enrichment statistic to compare a query signature (i.e., venoms) to all signatures in a reference database (i.e., known drugs), normalizing for cell lines and other confounding variables, and finally aggregating scores of ‘like’ signatures (i.e., drug mechanisms of action (MoAs)) using a maximum-quantile procedure. Complete details of these methods are provided in Section 4.7.1.

Different venoms yield different profiles of connectivity scores based on the genes present in their differential expression signatures. For example, all connectivity scores between *B. occitanus* and CMap perturbagens are zero, and all connectivity scores between *S. horrida* and CMap perturbagens are negative, which suggest that these venoms either behave like no known perturbagen classes, or that the venoms have no therapeutic activity on IMR-32 cells. Kernel density plots of the connectivity scores for each venom are shown in Figure 2. In Figure 3, we show several visualizations of the connectivity analysis results that highlight characteristics of the data. Interestingly, when hierarchical clustering is performed on the connectivity scores by venom perturbation, the venom perturbations form robust clustering patterns that persist across multiple non-overlapping subsets of the connectivity data. This suggests that the clustering corresponds to meaningful characteristics of the venom perturbations in comparison to known drugs, although these characteristics are not readily apparent (i.e., the clustering does not reproduce taxonomy or other obvious traits of the venoms). Similarities between perturbational expression signatures represent a similar function in biological systems (rather than the relatedness of the venoms’ species of origin). Therefore, the clustering patterns may indicate processes such as convergent evolution, which is widely acknowledged as a major driving factor in the development of venom arsenals [13].

### 2.4. Associations between Venoms and Existing Drugs

The associations we identified are shown in Table 4. As we anticipated, only some venoms show strong associations to any classes of drugs. Interestingly, only one venom (*S. subspinipes dehaani*) was linked to an ion channel inhibition MoA—venoms, in general, tend to have powerful ion-channel-blocking or -activating effects. However, this may be due to a preponderance of non-ion-channel MoAs in the CMap data rather than an actual lack of ability to identify ion channel activity.

Many of these MoAs comprise either well-established or emerging classes of cancer drugs. Some that have been used extensively as chemotherapeutic agents include CDK inhibitors (palbociclib, ribociclib, and abemaciclib), topoisomerase inhibitors (doxorubicin, teniposide, and irinotecan, among others), and DNA synthesis inhibitors (mitomycin C, fludarabine, and floxuridine). Meanwhile, PI3K inhibitors and FGFR inhibitors are classes of “emerging” chemotherapy drugs, each recently leading to many high-impact research studies and early-stage clinical trials. The other classes are indicated for a diverse range of diseases, including circulatory and mental conditions (calcium channel blockers) and cardiac abnormalities (ATPase inhibitors).

#### 2.4.1. Argiope Lobata Venom Versus Cardiopulmonary and Psychiatric Diseases

*A. lobata* is a species of spider in the same genus as the common garden spider. The species is relatively understudied, largely due to its lack of interaction with humans, despite being distributed across Africa and much of Europe and Asia. The venom from species of *Argiope* spiders contain toxins known as *argiotoxins* [15], which are harmless to humans, in spite of having inhibitory effects on AMPA, NMDA, kainite, and nicotinic acetylcholine receptors, which have been implicated in neurodegenerative and cardiac diseases. VenomSeq provides supporting evidence for therapeutic activity in each of these classes. Overall, the complete venom proteome of *A. lobata* is understudied, and the only component with significant existing research is argiotoxin-636 (sometimes referred to as argiopin), which is an inhibitor of AMPA receptors [16].

Connectivity analysis links *A. lobata* venom to ATPase inhibitor drugs (see Figure A6), which include digoxin, ouabain, cymarin, and other cardiac glycosides, and are used to treat a variety of heart conditions [17] and cancers [18]. Another venom-derived compound—bufalin (from the venom of toads in the genus *Bufo*) [19]—is considered an ATPase inhibitor, and has demonstrated powerful cardiotonic effects [20,21]. These compounds inhibit the cellular sodium–potassium ATPase ion channel, which has not previously been identified as a target of *A. lobata* venom components. However, other components isolated from the venom are known to inhibit the functionally similar glutamate receptor, making ATPase activity plausible [22]. Connectivity analysis also links the venom to PPAR agonist drugs, which are used to treat cholesterol disorders [23], diabetes [24], metabolic syndrome [25], and pulmonary inflammation [26]. Interestingly, PPARγ activation results in cellular protection from NMDA toxicity [27]. Given the known inhibitory effect of argiotoxins on NMDA receptors [28], this is striking and biologically plausible evidence for toxin synergism, where two or more venom components target multiple cellular structures with related functions in order to incite a more powerful response [29].

Master regulator analysis supports these findings, as well. We found that *A. lobata* venom is associated with a number of circulatory diseases, including hypertension, heart failure, cardiomegaly, myocardial ischemia, and others. Additionally, it reveals strong associations with an array of mental conditions, such as schizophrenia, bipolar disorder, and psychosis. These associations are supported by recent research into argiotoxins (and other polyamine toxins), showing that their affinity for iGlu receptors can be exploited to treat both psychiatric diseases and Alzheimer’s disease [15].

#### 2.4.2. Scorpio Maurus Venom for Cancer Treatment via FGFR Inhibition

*S. maurus*—the Israeli gold scorpion—is a species native to North Africa and the Middle East. Its venom is not harmful to humans, but it is known to contain a specific toxin, named maurotoxin, which blocks a number of types of voltage-gated potassium channels—an activity that is under investigation for the treatment of gastrointestinal motility disorders [30]. Existing proteomic analyses of *S. maurus* venom have identified 65 distinct components, which can be broadly categorized as antimicrobial peptides, insect-toxin-like peptides, sodium channel toxins, potassium channel toxins, and La1-like peptides (which have unclear function) [31]. Researchers have noted significant intraspecies variation in their venom proteomes based on geography [32].

Our connectivity analysis suggests an additional association with FGFR-inhibitor drugs. FGFR inhibitors are an emerging class of drugs with promising anticancer activity, and much research focused on them aims to understand and counteract their adverse effects (see Figure A7). Although there is no prior mention of FGFR-related activity from this or related species of scorpions, descriptions of unexpected side effects of *S. maurus* venom on mice provide evidence that such activity could be true. In particular, the venom has been shown to have biphasic effects on blood pressure: when injected, it causes rapid hypotension, followed by an extended period of hypertension. The fast hypotension is known to be caused by a phospholipase A${}_{2}$ in the venom, but no known components elicit hypertension when administered in purified form [33]. The observed FGFR-inhibitor-like effects on gene expression suggest that an unknown component (or group of components) may cause the hypertensive effect via FGFR inhibition.

### 2.5. Differential Expression Profiles of Purified Teretoxins

Above, we described applying VenomSeq to crude venoms comprising potentially many active ingredients, some of which may act synergistically. Another potential use of VenomSeq is to explore therapeutic potential of individual venom components, which tend to induce fewer gene expression changes and more closely resemble validated and approved pharmaceutical drugs (which usually consist of only one or a few active ingredients). Since individual venom components interact with fewer target structures and lack the synergistic effects found in crude venoms [34,35], it is important to determine whether VenomSeq has the sensitivity to detect the therapeutic effects of single components. To this end, we also constructed differential expression signatures for IMR-32 cells perturbed by nine purified teretoxin peptides. Of the nine teretoxins, four yielded statistically significant gene expression changes in IMR-32 cells. One of these, named Mki 8.7, produced a robust expression signature with 25 differentially expressed genes. All teretoxin expression signatures are available on FigShare at https://doi.org/10.6084/m9.figshare.22757963 (accessed on 6 July 2023).

### 2.6. Associations between Venoms and Disease Regulatory
Networks

Direct observations of expressed genes (via mRNA counts) provide an incomplete image of the regulatory mechanisms present in a cell. To complement the CMap approach that focuses on perturbations at the gene level, we designed a parallel approach that uses cell-regulatory network data to investigate perturbations at the regulatory module (e.g., pathways and metabolic networks) level; an approach we refer to as *master regulator analysis*. In master regulator analysis, the ARACNe algorithm [36] is used to obtain regulatory network data for our cell line of interest (in this case, IMR-32), consisting a list of regulons—overlapping sets of proteins whose expression is governed by a master regulator (e.g., a transcription factor). The msVIPER algorithm [37] is then used to determine the activity of each regulon by computing enrichment scores from observed expression levels of the genes/proteins contained in that regulon (here, using the RNA-Seq results described in Section 2.2).

We matched the significantly up- and downregulated master regulators for each venom to diseases using high-confidence TF–disease associations in DisGeNET [38]—a publicly available database of associations between diseases and gene network component. This approach is based on the idea that diseases caused by the dysregulation of metabolic and signaling networks can be treated by administering drugs that “reverse” the cause (i.e., abnormal master regulator activity) of dysregulation. Since we are interested in discovering associations with multiple corroborating pieces of evidence, we specifically filtered for diseases where *two or more* linked TFs are dysregulated when perturbed by the venom. The complete list of associations are provided on figshare at https://doi.org/10.6084/m9.figshare.7609793 (accessed on 6 July 2023); here, we describe a handful of interesting observations.

The most prevalent class of illness (comprising 19.7% of all associations across all venoms) is DISEASES OF THE NERVOUS SYSTEM AND SENSE ORGANS. This is not surprising, considering many of the 25 venoms have neurotoxic effects, and IMR-32 is a cell line derived from neuroblast cells. One source of bias in these results is that similar diseases tend to be associated with the same regulatory mechanisms [39]. For example, associations between a venom and schizophrenia will often be co-reported with associations to other mental conditions, such as bipolar disorder and alcoholism.

### 2.7. MOAs of Venoms versus Synthetic Small-Molecule Drugs

In the connectivity analysis portion of VenomSeq, we demonstrated that these techniques can identify novel venom–drug class associations and corroborate known venom activity. One distinct advantage of performing queries against the CMap reference dataset is their inclusion of manually curated perturbagen classes (PCLs), which allow for the normalization of data gathered from multiple perturbagens and multiple cell lines, aggregated at a class level that corresponds approximately with the drug mode of action. For this reason, hypotheses generated by the connectivity analysis portion of VenomSeq are often testable at the protein level.

One important caveat is that venom components have a tendency to interact with cell surface receptors (e.g., ion channels or GPCRs), inciting various signaling cascades and therefore acting indirectly on downstream therapeutic targets. While this is certainly the case for many drugs as well (GPCRs are considered the most heavily investigated class of drug targets [40]), small molecules can often be designed to enter the cell and interact directly with the downstream therapeutic target. This has important implications regarding assay selection for the in vitro validation of associations learned through the connectivity analysis. For example, if the MoA of interest is the inhibition of an intracellular protein (e.g., topoisomerase), a cell-based assay should be considered when testing venom hypotheses, since the venom likely is not interacting directly with the topoisomerase (and, therefore, the effect would not occur in non-cell-based assays).

### 2.8. Venoms Versus Human Diseases

The master regulator analysis portion of VenomSeq discovers associations between venoms and the diseases they may be able to treat, rather drugs. This could be especially useful for discovering treatments for diseases with no or few existing indicated drugs (or drugs that are not present in public differential expression databases). Additionally, since the master regulator approach is sensitive to complex metabolic network relationships, it is (theoretically) more sensitive to patterns, as well as more suited to diseases with complex genetic etiologies that are not explainable through observed gene counts alone.

Currently, the primary drawback to the master regulator approach is that criteria for statistical significance are not well established. Therefore, it is challenging to determine which venom–disease associations are most likely to reflect actual therapeutic efficacy. As a temporary alternative, we used several heuristics to ensure that there are multiple corroborating sources of evidence for the reported associations.

As discussed previously, the connectivity analysis produces hypotheses that are relatively straightforward to validate experimentally, using affordable, widely available assay kits and reagents. Since the master regulator workflow provides hypotheses at the disease level (where the underlying molecular etiologies can be unknown), validation instead needs to be performed at the *phenotype* level, either using animal models of disease or carefully engineered, cell-based phenotypic assays that measure response at multiple points in disease-related metabolic pathways (e.g., DiscoverX’s BioMAP^®^ platform [41]).

### 2.9. Specific Therapeutic
Hypotheses

VenomSeq leverages multiple modalities of data analysis for two reasons: (1) This allows us to cover diseases with a wider array of molecular etiologies, and (2) it provides a means for obtaining multiple pieces of corroborating evidence for a given hypothesis. If a link between a venom and a drug/disease is suggested by both connectivity analysis and master regulator analysis, and there is additional evidence in the literature that lends biological or clinical plausibility, this increases our confidence that the suggested therapeutic effect is robust.

Particularly regarding connectivity analysis and novel associations between venoms and existing drugs, it is important to remember that these associations are based on similarities in gene expression alone, which is a downstream cellular process that can result from interactions at various points in upstream signaling cascades. Therefore, saying that a venom’s effects on gene expression resemble those of a known PPAR agonist drug does not necessarily imply that the venom is itself a PPAR agonist. Nonetheless, even if the molecular mode of action is different, the therapeutic outcome may still be the same. For this reason, VenomSeq’s approach has a significant advantage over other drug discovery methods that focus solely on a prespecified molecular mode of action, which may have far-reaching implications in an era where novel drugs are limited by the number of currently known molecular targets (known as the ”druggable genome”) [42].

### 2.10. Transitioning from Venoms to Venom Components

VenomSeq is a technology for discovering early evidence that a *venom* has a certain therapeutic effect. However, most successful approved drugs derived from venoms make use of the activity of a single component within that venom, rather than the entire (crude) venom. As previously mentioned, venoms can comprise hundreds of unique components, each with a unique function and molecular target. Following this observation, we applied the VenomSeq pipeline to nine purified peptides from snails in the family Terebridae to assess whether VenomSeq can effectively produce differential expression profiles for individual venom components. We describe the resulting expression profiles in Section 2.5 and experimental methods in Section 4.3.

Of the nine teretoxins, five caused no significant changes in gene expression. This is consistent with our expectations—marine snail venoms components tend to have highly targeted modes of action, and any single cell line will respond to only some of the active components in a venom. Of the remaining four teretoxin peptides, one—named Mki8.7, from the venom of *Myurella kilburni*—produced a robust signature with 13 genes downregulated and 12 upregulated. We feel this merits further investigation, and typifies the type of workflow we would like to see used with VenomSeq in the future: Both crude venoms and individual venom components should be broadly screened for therapeutic effects, and in diverse human cell lines. Since isolating venoms and purifying their individual components is both laborious and expensive, a production-scale application of VenomSeq will be a costly endeavor, but one with significant potential for improving human health.

Furthermore, although most existing venom-derived drugs consist of a single component, crude venoms in nature use the synergistic effects of multiple components to cause specific phenotypic effects [29]. Therefore, testing each venom component individually using the VenomSeq workflow might fail to capture all of the clinically beneficial activities demonstrated by the crude venom. A brute-force solution is to perform VenomSeq on all combinations of the isolated venom components, but doing so requires a massive number of experiments ($2^{n}-1$, where *n* is the number of components in the venom). Therefore, it will be necessary to establish a protocol for prioritizing combinations of venom components. One potential solution is to fractionate the venom (i.e., using gel filtration) and perform VenomSeq on combinations of the fractions, but this will need to be tested. Alternatively, integrative systems biology techniques could be used to predict which components act synergistically, via similarity to structures with well-established activities.

### 2.11. Applying the VenomSeq Framework to Other Natural Product Classes

VenomSeq was designed for the purpose of discovering therapeutic activities from venoms, but it could be feasibly extended to other types of natural products, including plant and bacterial metabolites and immunologic components. Venoms provide a number of advantages and simplifying assumptions that were useful in designing the technology, but a broader application of VenomSeq will enable the relaxation of these assumptions with some minor modifications to experimental protocol and data analysis. For example, non-venom toxins may have less-targeted MoAs, disrupting biological systems indiscriminately (e.g., by interrupting cell membranes regardless of cell type). Additionally, the kinetics of non-venom natural products may be more subtle than venoms, which tend to have powerful binding and catalytic properties.

### 2.12. *VenomSeq* Technical Validation

Following the procedures described in Section 4.8, we used a secondary PLATE-Seq dataset of 37 existing drugs (with known effects) tested on IMR-32 cells to assess whether the sequencing technology (PLATE-Seq) and cell line (IMR-32) employed by VenomSeq are compatible with connectivity analysis and the CMap reference dataset. In this dataset, 20 of the 37 drugs have annotations to an existing CMap perturbational class (PCL). The drugs, their modes of action, and the PCLs of which they are members are listed in Table 5. Using these 20 drugs, we validated the performance of VenomSeq by demonstrating whether the ”true” PCL is assigned high connectivity scores both when using individual cell lines and when integrating scores across all available cell lines, as described below.

#### 2.12.1. VenomSeq Technical Validation: Recovering Connectivity by Integrating Cell Lines

When we aggregated all connectivity scores between a known drug and members of the same PCL in the CMap dataset, irrespective of cell line, the connectivity scores were significantly greater than those in a null model in 12 out of 20 instances, which indicates that drugs within the same functional class tend to have more similarities in the query and reference datasets than if the compounds are chosen at random. In all 20 cases, the average effect size (effect size is defined as the average difference between connectivities within the expected PCL and the null model of random connectivities for the same query) was positive, regardless of statistical significance. These—and their corresponding measures of significance—are shown in Figure 4 and Table 6. Overall, these data are congruent with those made by the Connectivity Map team in [14]—namely, that expected connections between query drugs and reference compounds can be recovered for some PCLs, but not for others. Importantly, in both our observations and the observations in [14], PCLs related to highly conserved core cellular functions perform better under this approach.

**Figure 4 toxins-15-00451-f004:**
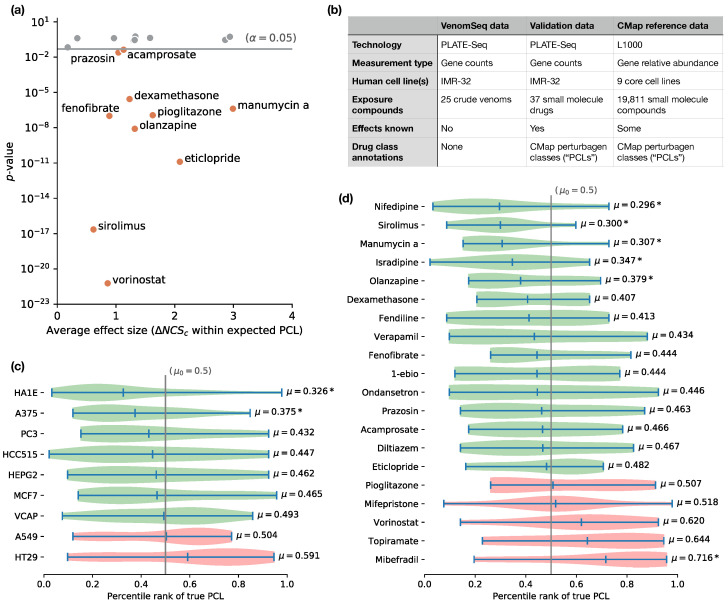
Results of applying the VenomSeq sequencing and connectivity analysis workflow to 37 existing drugs with known effects to validate the compatibility of PLATE-Seq and IMR-32 cells with the connectivity analysis algorithm and dataset. (**a**) Scatter plot showing validation drugs that are members of a CMap PCL and the mean differences between within-PCL connectivity scores and a null distribution of random connectivity scores for the same drug (Table 6). Vertical axis shows the *p*-value of a Student’s *t*-test comparing the within-PCL and null connectivity score distributions (corrected for multiple testing). Statistically significant drugs are labeled by name. (**b**) Summary of the validation strategy, showing that the validation dataset bridges certain gaps between the VenomSeq data and the CMap reference data. (**c**) Distributions of rank percentiles of expected (“true”) PCLs within the list of all PCLs ordered by average connectivity score (Table 7), aggregated by CMap dataset cell lines, and (**d**) validation drugs. Green distributions indicate a shift towards the front of the rank-ordered list, indicating stronger compatibility with the PLATE-Seq/IMR-32 query data, based on expected connections, and “*” indicates statistically significant shifts.

**Table 7 toxins-15-00451-t007:** Correct PCL ranks aggregated by cell line. Mean rank percentile is the mean rank of the correct (“true”) PCL, aggregated over all query drugs and divided by the total number of PCLs (92), reported by cell line.

CMap Cell Line	Mean Rank Percentile	FDR-Corrected *p*-Value
HA1E	0.326087	0.001663
A375	0.375000	0.004926
PC3	0.431522	0.109226
HCC515	0.446739	0.193877
HEPG2	0.461957	0.258068
MCF7	0.465217	0.279325
VCAP	0.492935	0.443995
A549	0.503804	0.468387
HT29	0.075445	0.591304

#### 2.12.2. VenomSeq Technical Validation: Impact of Reference Cell Lines and Query Drugs on Expected PCL Percentile Ranks

Since IMR-32 cells are not present in the CMap reference dataset, we were particularly interested in seeing which cell lines present in the reference dataset (if any) performed better than others at the task of recovering expected connections. Using the PCL ranking strategy described in Section 4.8, we found that seven of the nine core cell lines show at least a moderate tendency to place the true PCL towards the front of the ranked list of all PCLs, indicating that at least some of the ability to recover expected connections is retained when looking at those seven cell lines individually. PCL rankings stratified by drug (rather than cell line) show a similar pattern. In total, 15 of 20 PCL-annotated drugs tend to have the expected PCL ranked towards the front of the list (“enrichment”), while 5 tend to have the expected PCL show up towards the back of the list (“depletion”). It should be noted that—due to the rather small number of profiles in the reference dataset that are annotated to PCLs—these two analyses were limited in terms of statistical power, and deserve a follow-up analysis in the future, when more PCLs and members of those PCLs are present in the reference database.

#### 2.12.3. VenomSeq Technical Validation: Interpreting Connectivity Analysis Validation Results

In Section 2.12, we described the results of the connectivity analysis procedure applied to PLATE-Seq expression data from IMR-32 cells treated with 37 existing drugs that have known effects, many of which are members of Connectivity Map PCLs. Since VenomSeq uses an expression analysis technology that is different from the Connectivity Map’s L1000 platform, as well as a cell line that is not present in the Connectivity Map reference dataset, this is crucial for establishing that one can discover meaningful associations between crude venoms and profiles in the reference data within the VenomSeq framework.

Overall, the findings of our analysis are congruent with those made by the Connectivity Map team in [14]. Specifically, PCLs that affect highly conserved, core cellular functions (such as HDAC inhibitors, mTOR inhibitors, and PPAR receptors) tend to form strong connectivities with members of the same class regardless of cell line. Therefore, associations discovered between crude venoms and these drug classes are likely “true associations”, even when using IMR-32 cells in the analysis. Furthermore, by virtue of leveraging data corresponding to drugs with known effects, but using a new cell line and different assay technology, we have made the following novel findings:

Although IMR-32 is not present in the reference dataset, similarities between IMR-32 and cell lines that are present in the reference data can be leveraged to select reference expression profiles that are more likely to reproduce true associations. For example, HA1E and A375 cells produce expression profiles that form reasonably strong connectivities between IMR-32 query signatures and members of the same drug classes. More cell lines need to be included in the Connectivity Map data in order to better understand correlation structures in cell-specific expression, as well as to better capture therapeutic associations that are specific to cell types underrepresented in current datasets. It should also be noted that the nature of IMR-32 as an oncogenic cell line could contribute to false negatives or false positives in VenomSeq’s predicted associations, due to the disruption of certain key regulatory pathways, although it could also make the results more sensitive to therapeutic activities relating to cancer. Similarly, continued effort should be devoted to adding new PCL annotations. Currently, only 12.3% of compound signatures in the reference dataset are annotated to at least one PCL, and some PCLs contain only a few signatures. A more rigorous definition of what specifically comprises a PCL would allow secondary research groups to contribute to this effort, ultimately improving the utility of the CMap data and increasing the sensitivity of the algorithms used to discover new putative therapeutic associations.

In spite of the large amount of corroborating evidence these results provide (e.g., every drug in our validation set produced a positive average effect on within-PCL connectivities versus corresponding null distributions), we cannot definitively state whether the associations discovered for crude venoms reflect real therapeutic activities. Although our confidence in the novel associations would be improved by more PCL annotations to allow our analyses to attain greater statistical power, the ultimate test is to perform in vitro (and eventually in vivo) tests on individual venom components to detect these predicted therapeutic mechanisms of action. Initial cellular and protein-based assays suggest that the associations we found are real, but the toxicity of the crude venoms damages cells and membranes before the experiments can be run to completion. Aside from larger quantities of reference data against which to run the validation analyses, we also hope to employ other data science techniques involving network analysis and more advanced applications of master regulator analysis (see, e.g., Section 2.6) to further understand the dynamic interactions between cell types, gene expression, and perturbational signals that underlie therapeutic processes.

### 2.13. Accessing and Querying VenomSeq Data

VenomSeq is designed as a general and extensible platform for drug discovery, and we encourage the secondary use of both the technology as well as the data produced using the 25 venoms and 11 synthesized teretoxins tested on IMR-32 cells described in this manuscript. We maintain the data in two publicly accessible locations: (1) a “frozen” copy of the data, as it exists at the time of writing (on figshare, at https://doi.org/10.6084/m9.figshare.7611662) (accessed on 6 July 2023), and (2) a copy hosted on venomkb.org (accessed on 6 July 2023), available both graphically and programmatically, and designed to be expanded as new data and features are added to VenomKB.

## 3. Conclusions

Venoms provide an immensely valuable opportunity for drug discovery, but the enormous quantity and variety of compounds found in each venom arsenal requires a revision of the techniques used for identifying new therapeutic leads from venom natural products. Traditional methods—involving rigorous experimental validation and high cost—are necessary for validating associations between venoms and their respective therapeutic effects in living systems. However, data-driven computational approaches can make this process easier by generating new hypotheses backed by existing evidence and multiple levels of statistical validation. VenomSeq is an early example of such an approach.

VenomSeq takes a two-pronged approach, combining connectivity analysis and master regulator analysis to provide two orthogonal views of the effects venoms have on human cells, where likely therapeutic effects are validated using publicly available knowledge representations and databases. In this study, we tested the VenomSeq workflow on 25 diverse venoms and 9 purified terebrid venom components applied to human IMR-32 cells, and discovered a number of new therapeutic hypotheses supported by the existing literature evidence. In the overall scope of drug discovery from venoms, VenomSeq represents a scalable, generalizable, and novel technology for identifying lead compounds for subsequent experimental validation. As demonstrated above, it enables the discovery of new knowledge about venoms that are already well characterized, as well as for venoms about which relatively little is known. Since hypotheses are based on comparisons to existing drugs and diseases, the outcome is readily interpretable and can be applied to other classes of toxins or natural products.

To reinforce the validity of the hypotheses found through VenomSeq, future work is merited in applying the pipeline to new venoms and new human cell lines, and to test the pipeline on additional venoms, venom fractions, and isolated venom components with well-understood therapeutic modes of action. To fully bring a venom-based drug to market, it will also be necessary to experimentally validate the predictions (e.g., via both in vitro and animal-model-based approaches), optimize the compound for both stability and delivery, and progress through clinical trials to establish safety and efficacy. Although VenomSeq currently only addresses the portion of this pipeline involved in discovering new lead compounds, its extensibility represents an important step forward in venom-derived  therapeutics.

## 4. Materials and Methods

### 4.1. Reagents and Materials

We performed growth inhibition assays and perturbation experiments using IMR-32 cells—an adherent, metastatic neuroblastoma cell line used in previous applications of PLATE-Seq and VIPER—grown in FBS-supplemented Eagle’s Minimum Essential Medium (EMEM). Our use of the IMR-32 cell line is based on the fact that it is currently the only cell line compatible with the PLATE-Seq technology, as well as the availability of high-resolution regulatory network data that facilitate analysis using the VIPER algorithm (see Section 2.6). All venoms were provided in lyophilized form and stored at −20 °C. Since venoms naturally exist in aqueous solution, we reconstituted them in ddH${}_{2}$O at ambient temperature.

### 4.2. Obtaining 25 Venoms

VenomSeq is designed to apply to all venomous species across all taxonomic clades. Accordingly, we validated the workflow using 25 venoms sampled from a diverse range of species distributed across the tree of life. We selected the 25 species based on availability and compliance with international law, and sought to balance maximal cladistic diversity with minimal expected cytotoxicity (e.g., snakes in the genus *Bitis* are known for inducing tissue death and necrosis, and are therefore challenging to use for drug discovery applications [43]). We purchased the 25 venoms from Alpha Biotoxine in lyophilized form and obtained prior approval from the US Centers for Disease Control (CDC) through the Federal Select Agent Program [44] for importing venoms containing α-conotoxins. Generally, we sought to obtain 2–5 species each of snakes, spiders, scorpions, other arthropods, fish, amphibians, and mollusks. Due to the prevalence and ease of use of snake venoms, our final list includes 6 snakes. Other groups were selected largely based on commercial availability at the time of the study. The 25 venoms we selected are listed in Table 8. Note that we assigned a numeric identifier to each venom for convenience—these numbers show up numerous places in the data for VenomSeq. We also provide an evolutionary tree of the 25 species in Figure A1.

### 4.3. Obtaining 9 Purified Teretoxins

To assess the performance of VenomSeq on individual venom components, we selected 9 teretoxins from an in-house library of peptide sequences isolated from snails in the Terebridæfamily and synthesized purified samples using a method described previously [45].

### 4.4. Growth Inhibition Assays

A major challenge in generating differential gene expression data for discovery purposes is finding appropriate dosages for the compounds being tested. This is carried out to ensure the compound is in sufficient concentration to be exerting an observable effect on the cells, while also mitigating processes that result from toxicity (e.g., apoptosis). In practice, determining an appropriate dosage concentration usually makes use of previous experimental evidence and/or biochemical constants, but since these are generally not available for crude venoms, we instead determined dosages based on growth inhibition.

We prepared 2-fold serial dilutions of each venom, starting from 2.0 mg μL^−1^. We seeded 96-well plates with IMR-32 cells and exposed them to the serial dilutions of the venoms after 24 h of incubation. Then, 48 h after exposure, we quantified the growth inhibition of the IMR-32 cells via cell viability luminescence assays.

For each venom, we fit these data to the Hill equation:$$y=\mathrm{Bottom}+\frac{\left(\mathrm{Top}-\mathrm{Bottom}\right)}{1+10^{(\log{\mathrm{GI}}_{50}-x)\times h}}$$
where *x* is venom concentration, *y* is response (i.e., percent growth compared to untreated cells), Top and Bottom are the maximum and minimum values of *y*, respectively, and *h* is a constant that controls the shape of the sigmoidal curve. We used the resulting ${\mathrm{GI}}_{20}$ values (i.e., the value of *x* such that y=100%−20%=80%) as the venom exposure concentrations for the following sequencing experiments. We chose the ${\mathrm{GI}}_{20}$ metric based on previous experience in high-throughput screening studies when mechanistically based optimal dosages are not available—20% of growth inhibition tends to be a point at which the compound is noticeably exerting an effect on the cell, without gene expression being obscured by cell death or toxic processes. Since some of the curves had very steep slopes (indicating rapid loss of total cell viability after minuscule changes in venom concentration), we confirmed the accuracy of the ${\mathrm{GI}}_{20}$ concentrations via secondary viability assays using the exact ${\mathrm{GI}}_{20}$ values extrapolated from the growth inhibition curves.

### 4.5. mRNA Sequencing

We prepared samples of human IMR-32 cells in 96-well cell culture plates, allowing for 3 replicates at each of 3 time points (6, 24, and 36 h post-treatment) for each of the 25 venoms. The layout of the samples across two 96-well plates is available in Appendix B. We reconstituted the crude venoms in water and treated the samples with corresponding venoms at the previously determined ${\mathrm{GI}}_{20}$ values. We additionally prepared 12 control samples treated with water only and 9 control samples that were untreated. All samples for a venom were prepared from the same initial crude venom stock but incubated and processed individually prior to sequencing. To minimize the effect of microenvironments within the 96-well plate, samples for the same venom and time points were tiled across different regions of the plate. Following total mRNA extraction, we carried out the PLATE-Seq protocol [46] to obtain gene counts for each sample. All sequencing was performed on the Illumina HiSeq platform. We used STAR [47] to (1) map the demultiplexed, paired-end reads to the human genome (build GRCh38 [48], 3.1 Gb in length) and (2) count the reads uniquely mapping to known genes (e.g., when the maximum alignment score among alignments mapping to the correct strand is associated with a single gene). Complete details on the sequencing protocol are provided in the original PLATE-Seq publication [46]—deviations from the original protocol include the human genome assembly used (hg38 instead of hg19) and the aligner software used (STAR instead of bwa-mem). We used raw numbers of uniquely mapping counts to quantify the expression of each gene, as is appropriate for downstream analysis with the DESeq2 softare. For detailed quality control data for the sequencing experiments, refer to Appendix B.

### 4.6. Constructing Expression Signatures

We constructed differential gene expression signatures using the DESeq2 [49] library for the R programming language. DESeq2 fits observed counts for each gene to a negative binomial distribution with mean $\mu_{ij}$ and dispersion (variance) $\alpha_{i}$, which we find to be a more robust model than traditional approaches based on the Poisson distribution (i.e., by allowing for unequal means and dispersions). In practice, users can substitute any method for determining significantly up- and downregulated genes from count data. We filtered for genes with an FDR-corrected *p*-value <0.05 and recorded their respective mean $\log_{2}$-fold change values, noting whether expression increased (upregulated) or decreased (downregulated).

### 4.7. Comparing Venoms to Known Drugs and Diseases

Our overall procedure for comparing venoms to known drugs and diseases based on differential expression signatures is shown in Figure 5, and described below in further detail.

#### 4.7.1. Comparing to Known Drugs Using the Connectivity Map

We retrieved the most recently published Connectivity Map dataset from the Clue.io Data Library (GSE92742), which contains 473,647 perturbational signatures, each consisting of robust *Z*-scores for 12,328 genes, along with relevant metadata. We then used the procedure described by the Connectivity Map team [14] to generate connectivity scores between each of the VenomSeq gene expression signatures and each of the reference expression profiles in the Connectivity Map database. This procedure, adapted for VenomSeq, is summarized below.

Let a query $q_{i}$ be the two lists of up- and downregulated genes corresponding to the differential expression signature for venom *i*, and $r_{j}\in R$ be a vector of gene-wise *Z*-scores in reference expression signature *j*. We first generate a *Weighted Connectivity Score* (WCS) *w* between $q_{i}$ and

$r_{j}$:$$w_{qr}=\left\{\begin{array}{cc} (ES_{\mathrm{up}}^{q,r}-ES_{\mathrm{down}}^{q,r})/2 & \;\mathrm{if}\;\mathrm{sgn}\left(ES_{\mathrm{up}}^{q,r}\right)\neq \mathrm{sgn}\left(ES_{\mathrm{down}}^{q,r}\right)\\ 0 & \;\mathrm{otherwise} \end{array}\right.$$
where sgn denotes the sign function $\frac{d}{dx}\left|x\right|$, and $ES_{\;}^{qr}\cdot \;$ is the signed enrichment score for either the up- or downregulated genes in the signature, calculated separately (see below for details).

Although we validated VenomSeq on only a single human cell line, the reference database provided by the Connectivity Map provides expression profiles on 9 core cell lines across multiple classes of perturbagens. Therefore, we compute normalized versions of WCS called Normalized Connectivity Scores (NCSs):$$NCS_{q,r}=\left\{\begin{array}{cc} w_{q,r}/\mu_{c,t}^{+}\; & \mathrm{if}\;\mathrm{sgn}(w_{q,r})>0\\ w_{q,r}/\mu_{c,t}^{-}\; & \mathrm{otherwise} \end{array}\right.$$
where $\mu_{c,t}^{+}$ and $\mu_{c,t}^{-}$ are the means of all positive or negative WCSs (respectively) for the given cell line and perturbagen type.

The final step in computing connectivity scores between a venom *q* and a reference *r* is to convert $NCS_{q,r}$ into a value named τ, which represents the signed quantile score in the context of all positive or negative NCSs:$$\tau_{q,r}=\mathrm{sgn}\left(NCS_{q,r}\right)\frac{100}{N}\sum_{i=1}^{N}\left[|NCS_{i,r}|<|NCS_{i,r}|\right]$$
where *N* is the number of all expression signatures in the reference database and |NCS| is the absolute magnitude of an NCS.

##### Enrichment Score Computation

For a venom *q* and reference expression signature *r*, the enrichment score $ES_{\;}^{qr}\cdot \;$ is a signed Kolmogorov–Smirnov-like statistic indicating whether the subset of up- or downregulated genes in *q* tend to occur towards the beginning or the end of a list of all genes ranked by expression level in *r*. We follow a procedure similar to that described by Lamb et al. in [12]. Specifically, we compute the following two values:$$a=\max_{j=1}^{t}\left[\frac{j}{t}-\frac{V_{qr}\left(j\right)}{n}\right]$$
$$b=\max_{j=1}^{t}\left[\frac{V_{qr}\left(j\right)}{n}-\frac{\left(j-1\right)}{t}\right]$$
where $V_{qr}$ is the vector of non-negative integers that provides the indices of the genes in *q* within the list of all genes ordered corresponding to their assumed values in *r*, *t* is the number of genes in *q*, and *n* is the number of genes reported in the reference database (in practice, t≪n). We then set ES as follows:$$ES_{\;\cdot \;}^{qr}=\left\{\begin{array}{cc} a & \;\mathrm{if}\;a>b\\ -b & \;\mathrm{if}\;a<b \end{array}\right.$$

Since each query *q* consists of two lists—one of upregulated and one of downregulated genes—we compute both $ES_{\mathrm{up}}^{qr}$ and $ES_{\mathrm{down}}^{qr}$, respectively, and use these two values to compute $w_{qr}$, as described above.

#### 4.7.2. Comparing to Known Diseases Using Master Regulator
Analysis

We discovered associations between the venom expression profiles and known diseases (coded as UMLS concept IDs) as the result of two sequential steps: (1) algorithmic determination of substantially perturbed cell regulatory modules (called *regulons*), and (2) mapping master regulators to diseases using high-confidence associations distributed in the DisGeNET database. These took as input the same differential expression data used in the connectivity analysis. IMR-32 regulon data (in the form of an adjacency matrix, where nodes are genes and edges are measures of mutual information with respect to their coexpression) were provided by the authors of the ARACNe algorithm.

In order to identify perturbed regulons, we first performed a 2-tailed Student’s *t*-test between the genes’ expression in the ”test” set (samples perturbed by venoms) and the ”reference” set (control samples). To make the final expression signatures, we then converted the results of the *t*-tests to *Z*-scores, to make them consistent with the models used by downstream algorithms. We generated null scores by performing the same test on the expression data with permuted sample labels, to account for correlation structures between genes. Once we had computed *Z*-scores, we ran the msVIPER algorithm, which derives enrichment statistics for each regulon based on the expression levels of the genes contained in the regulon. The result of msVIPER is a table of regulons (labeled by their master regulator), with enrichment scores, *p*-values, and FDR-corrected adjusted *p*-values.

We then compared the significantly upregulated regulons to the manually curated subset of TF–disease associations from the DisGeNET database. To do so, we mapped the statistically significant master regulator TFs for each venom to TFs reported in DisGeNET, and then mapped those TFs to their associated diseases. To help with filtering venom–disease associations with low evidence, we only retained diseases where at least two of the regulons that were significantly dysregulated by the venom were associated with the same disease. Accordingly, we considered diseases with the highest number of significantly dysregulated master regulators to comprise the associations with the greatest amount of evidence.

Similarly to how we mapped drugs to drug classes, we mapped diseases to disease categories. To do so, we identified the set of ICD-9 codes for each disease, based on the diseases’ entries in the UMLS (UMLS CUIs were provided by DisGeNET). We then identified the disease category as the top-level ICD-9 “chapter” corresponding to that ICD-9 code (e.g., NEOPLASMS, MENTAL DISORDERS, DISEASES OF THE RESPIRATORY SYSTEM, etc.). In rare instances where a disease or condition was present in two locations (e.g., ”hypertension” is found in 2 chapters: DISEASES OF THE CIRCULATORY SYSTEM (401) and INJURY AND POISONING (997.91)), we opted for the more specific of the two (e.g., avoiding entries containing “not elsewhere classified”).

### 4.8. Assessing Sequencing Technology and Cell Type
Compatibility

Since VenomSeq uses a sequencing technology (PLATE-Seq) and a cell line (IMR-32) that have not been used previously with the connectivity analysis approach, we evaluated their compatibility using a secondary dataset consisting of IMR-32 cells perturbed with 37 drugs and sequenced using PLATE-Seq. Since these drugs have known effects—and since many are present in the L1000 reference dataset—we sought to determine the extent to which connectivity analysis captures functional similarities between these drug data and the L1000 reference expression profiles. The 37 drugs are listed in Table 5. For the purposes of this discussion, a “query signature” is an expression signature corresponding to one of the 37 drugs in the validation dataset, and a “reference profile” is an L1000 expression profile from the dataset (GSE92742) published by the Connectivity Map team and used in the crude venom connectivity analysis.

Using these data (consisting of gene count matrices with several technical replicates per drug), we constructed differential expression signatures and performed the connectivity analysis algorithm in the same manner as we had for IMR-32 cells exposed to the 25 crude venoms. We annotated each of the 37 drugs (where possible) with perturbagen classes (PCLs) defined by the Connectivity Map team, which allowed us to identify L1000 expression profiles that come from the same drug classes as the drugs in our validation dataset. We then evaluated connectivity scores among members of the same PCL from two perspectives: (1) By aggregating all τ scores for reference profiles corresponding to a given compound, integrating evidence from all cell lines, and (2) by aggregating τ scores within individual cell lines, allowing us to assess the degrees to which specific cell lines are compatible with IMR-32/PLATE-Seq query signatures.

For the first of these two approaches, we collected all values of τ connecting query signatures in a PCL to reference profiles in the same PCL, and constructed null models by retrieving τ scores between the same query signature and all reference profiles that are members of any PCL. We defined the “effect size” of each PCL annotation as the difference in the mean of the scores within the true PCL and the mean of the scores in the null model. Additionally, we determined statistical significance using independent two-sample Student’s *t*-tests. To correct for multiple testing, we adjusted *p*-values using the Benjamini–Hochberg procedure (α=0.05).

For the second approach—in which we evaluated each of the 9 core L1000 cell lines separately for each query signature—we retrieved τ scores between query signatures and each of the 92 PCLs in the reference dataset. Then, for each of the 9 cell lines and each of the query signatures annotated to a PCL, we constructed ordered lists of all PCLs ranked by their mean τ score in descending order (highest to lowest connectivity). In each of those lists, we determined the rank corresponding to the expected (“true”) PCL—which we call the *rank percentiles*—and aggregated these ranks separately by (a) the drug corresponding to the query signature and (b) cell line of the reference profile. These two strategies allow us to separately assess the effects of *drugs* and *cell lines* on the behavior of connectivity scores. Under the null hypothesis that there is no selective preference for the true PCL in the connectivity data, the mean rank percentiles would follow a continuous uniform distribution in the range [0,1]. Alternatively, if there is a selective preference for the expected PCL in the connectivity data, this rank will tend to occur towards the front of the list of ranks (and vice versa).

## Figures and Tables

**Figure 1 toxins-15-00451-f001:**
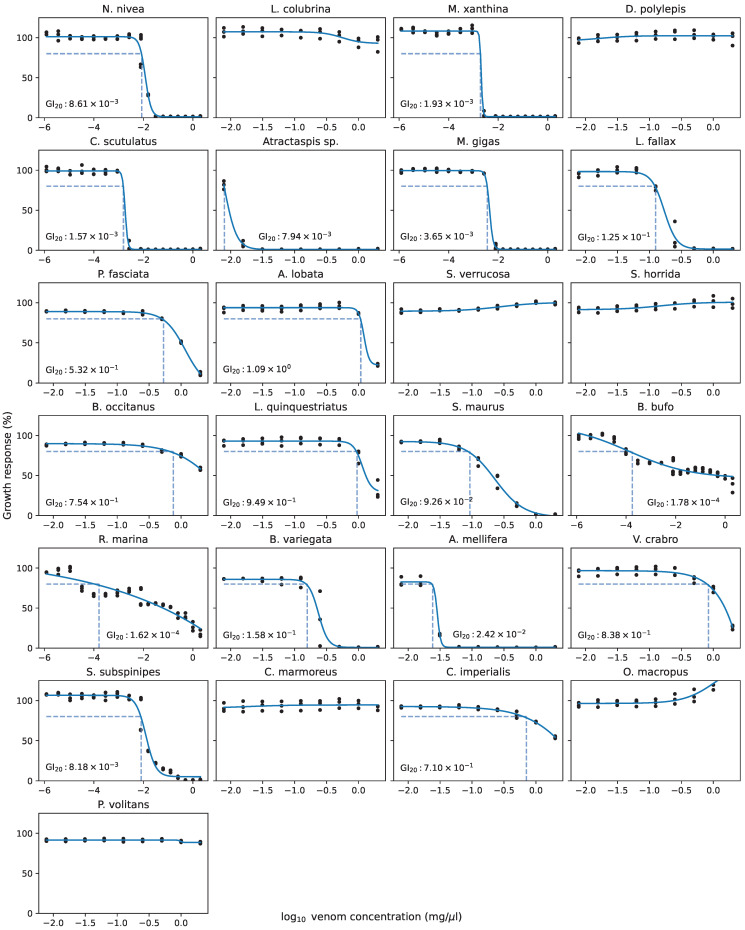
Growth inhibition plots for each of the 25 venoms. ${\mathrm{GI}}_{20}$ values are provided, unless growth inhibition was not observed (in which case sequencing was instead performed at 2 mg μL^−1^). Dashed lines indicate x- and y-intercepts of ${\mathrm{GI}}_{20}$ values.

**Figure 2 toxins-15-00451-f002:**
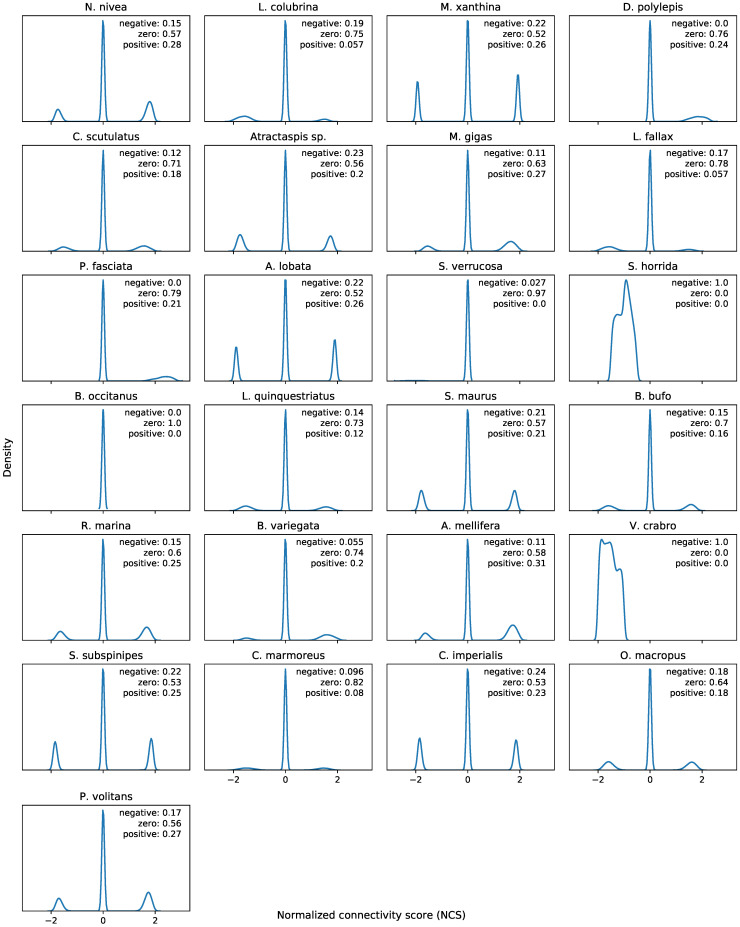
Kernel density plots of normalized connectivity scores (NCSs) for each of the 25 venoms. Note the tendency to introduce sparsity by setting NCS to zero if the quantities *a* and *b* have opposite signs (see Section 4.7.1). Text labels indicate proportion of NCSs for a single venom that are negative, zero, or positive. Each plot is based on 473,647 NCSs (all differential expression profiles in GSE92742 [14]).

**Figure 3 toxins-15-00451-f003:**
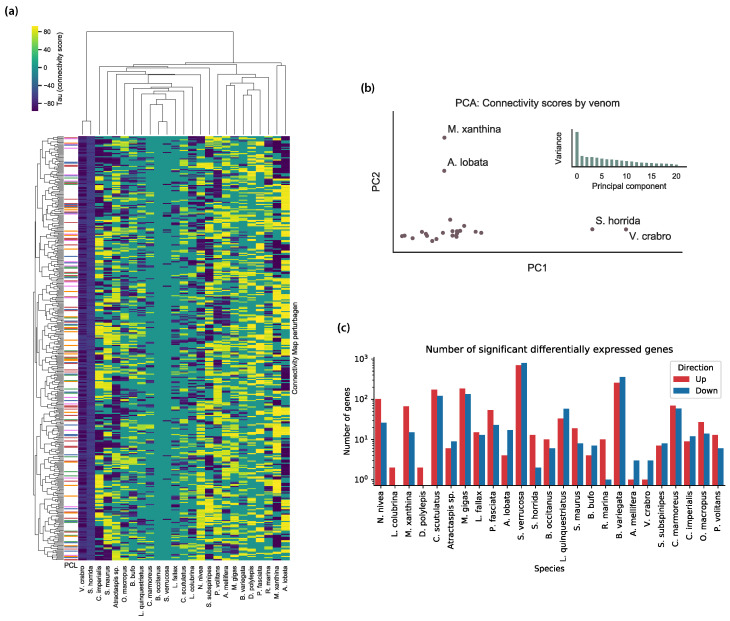
Connectivity analysis results. (**a**) Heatmap of τ-scores between the 25 venom perturbations and the 500 Connectivity Map signatures with the highest variance across all venoms. A distinct hierarchical clustering pattern is evident across the venom perturbations, although it does not conform to any obvious grouping pattern of the venoms. (**b**) Principle component analysis of the 25 venom perturbations, where features are all τ-scores between the venom and signatures from the Connectivity Map reference database. Four distinct outliers are labeled—these venoms correspond to outliers in the heatmap. Also shown are the ratios of variance explained by each of the first 21 principle components—after the first principle component, the distribution is characterized by a long tail, suggesting that much of the variance is spread across many dimensions, underscoring the complexity of the connectivity score data. (**c**) Barplot showing the number of significant differentially expressed genes for IMR-32 cells exposed to each of the 25 venoms.

**Figure 5 toxins-15-00451-f005:**
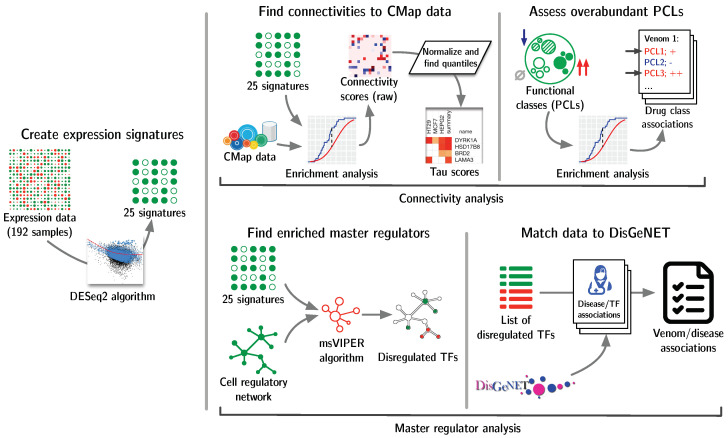
Strategy for discovering new associations from VenomSeq data. After obtaining processed gene counts per sample, we generated differential expression signatures for each venom and then used the signatures in two parallel analyses: connectivity analysis and master regulator analysis.

**Table 1 toxins-15-00451-t001:** Statistics for *S. maurus* growth inhibition data.

*S. maurus* Venom vs. IMR-32		
${\mathrm{GI}}_{20}$ (μg μL^−1^)		0.0926
$R^{2}$		0.991
Hill slope	Bottom	−2.096
	Top	92.572
	$\log{\mathrm{GI}}_{50}$	−0.640
	Slope (*h*)	−1.928

**Table 2 toxins-15-00451-t002:** Experimental conditions for RNA-Seq.

Venoms	25 species
Cell line	IMR-32 (Human neuroblastoma)
Dosage	${\mathrm{GI}}_{20}$ for each venom
Time points	6/24/36 h post-treatment
Replicates	3 per time point per venom
Controls	12 water controls, 9 untreated
Solvent	Water

**Table 3 toxins-15-00451-t003:** Partial differential expression signature for *O. macropus*. Most of the significantly differentially expressed genes (35 of 41 total) are omitted for brevity.

Gene	Base Mean	$\log_{2}$-FC	Wald Statistic	*p*-Adj
SPRY4	37.38	−2.27534	−3.3084	0.0991
REPIN1	38.30	−0.95256	−4.3326	0.0061
DUSP14	33.88	−0.91311	−3.3327	0.0991
⋮	⋮	⋮	⋮	⋮
BRD3	130.81	1.37645	4.115	0.0096
RSRC1	63.48	1.38140	4.2042	0.0091
BAZ1B	120.05	1.69463	5.0846	0.0003

**Table 4 toxins-15-00451-t004:** Venom–drug class associations.

Venom	Drug Class (MoA)
*Synanceia horrida*	ATPase inhibitor
	CDK inhibitor
	DNA synthesis inhibitor
*Scolopendra subspinipes dehaani*	T-type Ca${}^{2+}$ channel inhibitor
*Pterois volitans*	Topoisomerase inhibitor
*Argiope lobata*	ATPase inhibitor
	PI3K inhibitor
	PPARγ agonist
*Scorpio maurus*	FGFR inhibitor
*Rhinella marina*	HIV protease inhibitor

**Table 5 toxins-15-00451-t005:** Drugs used to validate PLATE-Seq and the IMR-32 cell line for connectivity analysis. Not all compounds of a given mechanism of action will necessarily map to that mechanism’s associated PCL—PCLs consist of compounds that are members of the same functional class and also have high transcriptional impact.

Drug	Mechanism of Action	CMap Perturbagen Class (PCL)
Mibefradil	T-type Ca${}^{2+}$ channel inhibitor	CP_T_TYPE_CALCIUM_CHANNEL_BLOCKER
Isradipine	L-type Ca${}^{2+}$ channel inhibitor	CP_CALCIUM_CHANNEL_BLOCKER
Nifedipine	L-type Ca${}^{2+}$ channel inhibitor	CP_CALCIUM_CHANNEL_BLOCKER
Diltiazem	Ca${}^{2+}$ channel inhibitor	CP_CALCIUM_CHANNEL_BLOCKER
Verapamil	Ca${}^{2+}$ channel inhibitor	CP_CALCIUM_CHANNEL_BLOCKER
Fendiline	Ca${}^{2+}$ channel inhibitor	CP_CALCIUM_CHANNEL_BLOCKER
Topiramate	Na${}^{+}$ and Ca${}^{2+}$ channel modulator	CP_SODIUM_CHANNEL_BLOCKER
Ionomycin	Ca${}^{2+}$ channel signal inducer	
1-EBIO	Ca${}^{2+}$-gated K${}^{+}$ channel activator	CP_POTASSIUM_CHANNEL_ACTIVATOR
Forskolin	Adenylyl cyclase activator	
Pregabalin	Increases GABA biosynthesis	
Gabapentin	Increases GABA biosynthesis	
Baclofen	GABA${}_{B}$-receptor agonist	
Memantine	Glu-receptor inhibitor	
Acamprostate	Glu-receptor inhibitor	CP_GABA_RECEPTOR_ANTAGONIST
MTEP	Glu-receptor inhibitor	
Ivermectin	Glu-gated Cl${}^{-}$ channel inhibitor	
Carbenoxolone	Glucocorticoid metabolism inhibitor	
Mifepristone	Glucocorticoid receptor inhibitor	CP_PROGESTERONE_RECEPTOR_ANTAGONIST
Dexamethasone	Glucocorticoid receptor agonist	CP_GLUCOCORTICOID_RECEPTOR_AGONIST
Aldosterone	Mineralocorticoid receptor agonist	
Spironolactone	Mineralocorticoid receptor inhibitor	
Olanzapine	Dopamine receptor inhibitor	CP_DOPAMINE_RECEPTOR_ANTAGONIST
Eticlopride	Dopamine receptor inhibitor	CP_DOPAMINE_RECEPTOR_ANTAGONIST
Ondansetron	5-HT${}_{3}$ serotonin receptor inhibitor	CP_SEROTONIN_RECEPTOR_AGONIST
Naltrexone	Opioid receptor inhibitor	
Disulfiram	Acetaldehyde dehydrogenase inhibitor	
Cerlitinib	ALK inhibitor	
Crizotinib	ALK inhibitor	
Sirolimus	mTOR inhibitor	CP_MTOR_INHIBITOR
Manumycin A	Farnesyltransferase inhibitor	CP_NFKB_PATHWAY_INHIBITOR
Vorinostat	HDAC (I/II/IV) inhibitor	CP_HDAC_INHIBITOR
Prazosin	Adrenergic receptor inhibitor	CP_BETA_ADRENERGIC_RECEPTOR_AGONIST
Rolipram	Phosphodiesterase-4 inhibitor	
Minocycline	NOS inhibitor	
Pioglitazone	PPARγ/α inhibitor	CP_PPAR_RECEPTOR_AGONIST
Fenofibrate	PPARα agonist	CP_PPAR_RECEPTOR_AGONIST

**Table 6 toxins-15-00451-t006:** Enrichment of strong connections in expected PCL annotations. *p*-values correspond to independent, two-sample Student’s *t*-tests between “within-PCL” connectivities and a null model of randomly sampled compound connectivities (see text) for the same query drug, and are corrected for multiple testing using the Benjamini–Hochberg procedure. Bolded *p*-values indicate statistical significance after correcting for multiple testing. Effect size is the difference in means between those two groups, such that larger effect sizes correspond to higher expected connectivity scores between the query drug and members of its same drug class. Note that effect sizes are relatively small in most cases—this is due in part to the sparsity of connectivity scores.

Drug	PCL	*p*-Value	Effect Size
Topiramate	CP_SODIUM_CHANNEL_BLOCKER	**1.018 × 10 ${}^{-31}$**	13.168
Vorinostat	CP_HDAC_INHIBITOR	**5.952 × 10 ${}^{-22}$**	1.717
Sirolimus	CP_MTOR_INHIBITOR	**2.240 × 10 ${}^{-17}$**	1.232
Eticlopride	CP_DOPAMINE_RECEPTOR_ANTAGONIST	**1.278 × 10 ${}^{-11}$**	4.175
Olanzapine	CP_DOPAMINE_RECEPTOR_ANTAGONIST	**8.117 × 10 ${}^{-9}$**	2.640
Fenofibrate	CP_PPAR_RECEPTOR_AGONIST	**1.012 × 10 ${}^{-7}$**	1.775
Pioglitazone	CP_PPAR_RECEPTOR_AGONIST	**1.158 × 10 ${}^{-7}$**	3.252
Manumycin A	CP_NFKB_PATHWAY_INHIBITOR	**4.124 × 10 ${}^{-7}$**	5.983
Dexamethasone	CP_GLUCOCORTICOID_RECEPTOR_AGONIST	**2.741 × 10 ${}^{-6}$**	2.462
Prazosin	CP_BETA_ADRENERGIC_RECEPTOR_AGONIST	**2.476 × 10 ${}^{-2}$**	2.083
Acamprosate	CP_GABA_RECEPTOR_ANTAGONIST	**4.290 × 10 ${}^{-2}$**	2.260
Mibefradil	CP_T_TYPE_CALCIUM_CHANNEL_BLOCKER	6.871 × 10 ${}^{-2}$	0.355
1-EBIO	CP_POTASSIUM_CHANNEL_ACTIVATOR	2.573 × 10 ${}^{-1}$	2.597
Fendiline	CP_CALCIUM_CHANNEL_BLOCKER	2.854 × 10 ${}^{-1}$	2.636
Diltiazem	CP_CALCIUM_CHANNEL_BLOCKER	2.929 × 10 ${}^{-1}$	5.719
Isradipine	CP_CALCIUM_CHANNEL_BLOCKER	4.062 × 10 ${}^{-1}$	0.683
Nifedipine	CP_CALCIUM_CHANNEL_BLOCKER	4.100 × 10 ${}^{-1}$	1.932
Mifepristone	CP_PROGESTERONE_RECEPTOR_ANTAGONIST	4.309 × 10 ${}^{-1}$	3.160
Verapamil	CP_CALCIUM_CHANNEL_BLOCKER	5.404 × 10 ${}^{-1}$	5.880
Ondansetron	CP_SEROTONIN_RECEPTOR_AGONIST	5.710 × 10 ${}^{-1}$	2.659

**Table 8 toxins-15-00451-t008:** The 25 venoms used to validate the VenomSeq workflow. Numbers in the right column are used as placeholder names for the venoms in data files.

Species Name	Common Name	Venom Number
*Naja nivea*	Cape cobra	1
*Laticauda colubrina*	Banded sea krait	2
*Montivipera xanthina*	Ottoman viper	3
*Dendroaspis polylepis polylepis*	Black mamba	4
*Crotalus scutulatus scutulatus*	Mojave rattlesnake	5
*Atractaspis* sp.	Burrowing asp	6
*Macrothele gigas*	Japanese funnel web spider	7
*Linothele fallax*	Tiger spider	8
*Poecilotheria fasciata*	Sri Lanka ornamental spider	9
*Argiope lobata*	-	10
*Synanceia verrucosa*	Reef stonefish	11
*Synanceia horrida*	Estuarine stonefish	12
*Buthus occitanus*	Common yellow scorpion	13
*Leiurus quinquestriatus*	Deathstalker	14
*Scorpio maurus*	Large-clawed scorpion	15
*Bufo bufo*	Common toad	16
*Rhinella marina*	Cane toad	17
*Bombina variegata*	Yellow-bellied toad	18
*Apis mellifera*	Western honey bee	19
*Vespa crabro*	European hornet	20
*Scolopendra subspinipes dehaani*	Vietnamese centipede	21
*Conus marmoreus*	Marbled cone snail	22
*Conus imperialis*	Imperial cone snail	23
*Octopus macropus*	Atlantic white-spotted octopus	24
*Pterois volitans*	Red lionfish	25

## Data Availability

All data related to this study are freely and publicly available. Code for VenomSeq can be found on Github at https://github.com/jdromano2/venomseq (accessed on 6 July 2023), and in archived form on Zenodo at https://doi.org/10.5281/zenodo.7897068 (accessed on 6 July 2023). Raw PLATE-Seq data are available on NCBI’s GEO database under accession GSE126575. All other data—including the processed results of connectivity analyses, technical validation studies, and others—are available on the VenomKB website (https://venomkb.org/ (accessed on 6 July 2023) and on FigShare at https://figshare.com/projects/VenomSeq/57425 (accessed on 6 July 2023).

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
