# Peer review of "Discovering Venom-Derived Drug Candidates Using Differential Gene Expression"

_toxins, 2023, doi:10.3390/toxins15070451_

Round 1

Reviewer 1 Report

Authors described VenomSeq in the therapeutic discovery of venom components of 25 species and purified toxins. The manuscript has some merits, but some clarifications are required to improve the manuscript. 

1. Introduction - Please elaborate on the problems/research gaps that lead to the current study's design. Provide examples of drug discovery from venoms, are they discovered through high-throughput screening? 

2. Methods - authors must explain the selection criteria (or why did you choose the 25 species?) of the venoms in this study; is this only restricted to the choice that the venoms have minimal cytotoxicity? In this case, are authors looking at anti-proliferative (or perhaps anti-cancer) drug discovery from venoms? 
This leads to my next doubts.

3. Venom components can be hemotoxic, cytotoxic, or even neurotoxic - why are authors focusing only on the growth inhibition of human cells? 

4. In Hiseq sequencing of genes, describe the sequencing parameters (how many reads, pair ends etc.); then explain how big is the human genome data used for blasting; how did you perform the quantification of the genes, i.e. is it by FPKM? 

5. The work involved a lot of bioinformatic analyses, is it very important to validate the outcomes from the bioinformatic analyses, have you considered/performed such validation studies?

6. In the results, authors showed 'a distinct hierarchical clustering pattern across venom perturbations', but there is no obvious grouping in the venom. This is quite unlikely because a diversity of venom compositions shall be observed. What is your possible hypothesis for this phenomenon? 

7. By mapping the RNAseq expression data to MOA (drug class), authors have identified multiple potential targets, e.g. on cell surface ion channels and intracellular proteome targets. What are the main conclusions to be drawn here? Can you also classify the MOA based on venom species? And, correlate with the venom compositions of each venom. 

8. Conclusion - Elaborate on how applicable is VenomSeq in drug discovery from venoms. 

Only minor errors, overall the language is easy to understand. 

Reviewer 2 Report

1. Can the authors explain this in the text:
a. Why these specific 25 species?
b. Why did the authors choose to use a GI of 20%? Why not 15%? 25%?

2. Table 2... Are these biological triplicates or just 3 aliquots from each time point? The former is preferred.

3. The authors should combine Results and Discussion, and edit accordingly.

4. More references should be included regarding what venom and therapeutics have in common to support your Discussion and Conclusions.

Reviewer 3 Report

The authors have introduced a new platform called VenomSeq, designed to identify venoms with therapeutic potential through the analysis of differential gene expression in human cell lines treated with various types of venoms. This platform also incorporates connectivity analyses, which involve comparing the differential expression profiles with registered data for known drugs and disease regulatory networks. Although the chosen cell line lacks reference data, the authors have provided several validation procedures that support the utility of the presented platform. The paper is intriguing, and the generated hypotheses, as well as the limitations of VenomSeq, are appropriately addressed and discussed.

Overall, the paper is well-structured and well-written. Two minor points to note are that tables and figures should be positioned below the corresponding text, and there are a few unnecessary hyphens scattered throughout the text.

Reviewer 4 Report

The authors of the manuscript "Discovering venom-derived drug candidates using differential gene expression" have performed an extensive analysis to identify candidate drug-like components in the venoms from different sources by implementing RNA-seq technology. The idea behind the study holds good promises in identifying target venom proteins/peptides of clinical importance. However, there are a few major concerns that need to be addressed for a proper understanding of the manuscript for the readers:

1. The authors have failed to justify the use of IMR-32 cell line for this experiment. It is important to note that IMR-32 being a oncogenic cell line has several dysregulated pathways that may or may not interfere with the effects of venom proteins/peptides.

2. The authors should also include a control primary cell line (such as HEK293, myocytes, etc.) in the study to understand and check the basal effect of venoms on different physiological pathways.

3. The authors have mentioned in their discussion that a majority of the effects shown by venom treatment is a combinatorial effect of several toxins, and the same toxins in isolation may fail to demonstrate a similar phenotypic effect. Therefore, a brief description of the proteome/metabolome for each venom on table 4 would give better insights in understanding the relationships between the venom source and the drug class.

4. The extrapolated correlation between the venom effects vs commercial drugs is to some extent overstretched. It is important to keep in mind that small molecule drug candidates significantly differ in the mode of action as compared to macromolecules (proteins and peptides) in venom.

5. The correlations drawn from the RNA-seq data should also be selectively reconfirmed using other in vitro approaches (qPCR, western blotting, etc). 

Minor comment:

1. Authors are suggested to revise the manuscript by using better scientific terminologies for the convenience of the readers. Eg. 'human cells' can be replaced with the exact cell type being used in the study.

2. Abbreviations and scientific names should be elaborated in their first mention. 

Reviewer 5 Report

This MS desribes the VenomSeq pipeline, which merges experimental data about the changes in gene expressions caused by the application of different venom compounds with similar data from known diseases / drugs to target venoms with a potential therapeutic application. As such, it has an obvious utility in trying to filter out only those venoms with the potential to be good drug candidates.

As interesting and pontentially useful as the idea is, the MS unfortunately does not do it justice.

First, I find the MS extremely difficult to read. The style is choppy and the points are not well integrated. The Resuls section in particular seems to be a collection of random sections at times with little flow or logical connection between them. In addition, the MS contains huge amounts of complex data that are not explained very well and sometimes seem to be included more to impress than to convince.

What I could glean from the MS is that VenomSeq appears to have a high false-negative error rate, and possibly an unexceptably high one. As pointed out in the MS in L129, the assumed major mode of action of most venoms could not be recovered here because these data are lacking in the reference database. In addition, VenomSeq only had about a 50% success rate in characterizing known drugs. Together, these characteristics severely limit the ability of VenomSeq to efficiently target drug candidates and especially those with a novel mode of action.

More minor or specific comments include:

L28: The first line of the sentence is unsupported by the rest of the paragraph. Just because one venom has hundreds of components doesn't make venoms (or even that venom) diverse. The same is true for the number of venomous animals unless they possess different venoms.

L31: "evolutionarily optimized" is meaningless jargon. What in nature isn't evolutionarily optimized?

L50: It would be helpful to briefly state what the goal of this comparison here is. I presume that VenomSeq looks for venom perturbation profiles that are the mirror image of disease ones.

L53: The Introduction needs to end with with a much better seque to what the rest of the paper is about, especially given that the M&M is shunted toward the end of the MS. The experimental results on the next page literally come out of nowhere and there is no real context to understand them or their purpose.

L53: Is Fig. 1 really necessary? The workflow is not that complex.

L62: Although the text here talks about GI20 values, GI80 values are given in the Fig. 2.

L93: I do not understand what role the teretoxin analyses here play. VenomSeq appears to try to correlate changes in gene expression, regardless of how they are induced. Does it really matter if these changes on the venom side come from a single peptide or a cocktail of them? Please justify.

L102: In the last paragraph of the Introduction, the comparison was with the gene-expression profiles of diseases, not drugs.

L147: As important is that the drugs map to the *correct* perturbational class.

L419: Same as with Fig. 1: is Fig. 6 really all that necessary?

Fig. 7: This is an unrooted tree and definitely not a cladogram. Either the source of the tree or how it was generated needs to be supplied.

The English is fine, but I find the MS to be extremely choppy and therefore difficult to read and understand.

Round 2

Reviewer 1 Report

The manuscript has been exceptionally improved. 

Do check for minor grammatical error. 

Reviewer 2 Report

The authors have fully answered my concerns.

There is the issue with biological triplicates... this means from 3 different/separate samples. The authors do mention that the samples were pooled. Given the caliber of the journal and the revised manuscript, however, this is acceptable.

Reviewer 4 Report

The concerns were well addressed. The revised manuscript can be accepted in the present form.